# A Tentative Study of the Effects of Heat-Inactivation of the Probiotic Strain *Shewanella putrefaciens* Ppd11 on Senegalese Sole (*Solea senegalensis*) Intestinal Microbiota and Immune Response

**DOI:** 10.3390/microorganisms9040808

**Published:** 2021-04-12

**Authors:** Marta Domínguez-Maqueda, Isabel M. Cerezo, Silvana Teresa Tapia-Paniagua, Inés García De La Banda, Xabier Moreno-Ventas, Miguel Ángel Moriñigo, Maria Carmen Balebona

**Affiliations:** 1Departamento de Química Inorgánica, Cristalografía y Mineralogía, Facultad de Ciencias, Campus de Teatinos s/n, Universidad de Málaga, 29071 Málaga, Spain; b22ceori@uma.es (I.M.C.); stapia@uma.es (S.T.T.-P.); morinigo@uma.es (M.Á.M.); balebona@uma.es (M.C.B.); 2Spanish Institute of Oceanography, Oceanographic Center of Santander, 39080 Santander, Spain; ines.gbanda@st.ieo.es; 3Ecological Area of Water and Environmental Sciences and Technics, University of Cantabria, 39005 Santander, Spain; morenox@unican.es

**Keywords:** inactivated probiotics, *Solea senegalensis*, *Shewanella putrefaciens* Ppd11, immune response, microbiota

## Abstract

Concerns about safety, applicability and functionality associated with live probiotic cells have led to consideration of the use of non-viable microorganisms, known as paraprobiotics. The present study evaluated the effects of dietary administration of heat-inactivated cells of the probiotic strain *Shewanella putrefaciens* Ppd11 on the intestinal microbiota and immune gene transcription in *Solea senegalensis*. Results obtained were evaluated and compared to those described after feeding with viable Pdp11 cells. *S. senegalensis* specimens were fed with basal (control) diet or supplemented with live or heat inactivated (60 °C, 1 h) probiotics diets for 45 days. Growth improvement was observed in the group receiving live probiotics compared to the control group, but not after feeding with a probiotic heat-inactivated diet. Regarding immune gene transcription, no changes were observed for *tnfα*, *il-6*, *lys-c1*, *c7*, *hsp70*, and *hsp90aa* in the intestinal samples based on the diet. On the contrary, *hsp90ab*, *gp96*, *cd4*, *cd8*, *il-1β*, and *c3* transcription were modulated after probiotic supplementation, though no differences between viable and heat-inactivated probiotic supplemented diets were observed. Modulation of intestinal microbiota showed remarkable differences based on the viability of the probiotics. Thus, higher diversity in fish fed with live probiotic cells, jointly with increased *Mycoplasmataceae* and *Spirochaetaceae* to the detriment of *Brevinemataceae*, was detected. However, microbiota of fish receiving heat-inactivated probiotic cells showed decreased *Mycoplasmataceae* and increased *Brevinemataceae* and *Vibrio* genus abundance. In short, the results obtained indicate that the viable state of Pdp11 probiotic cells affects growth performance and modulation of *S. senegalensis* intestinal microbiota. On the contrary, minor changes were detected in the intestinal immune response, being similar for fish receiving both, viable and inactivated probiotic cell supplemented diets, when compared to the control diet.

## 1. Introduction

Aquaculture is the fastest growing agro-industry sector in the world [1]. However, intensive aquaculture practices negatively affect the farmed fish physiology, disrupting the immune status and making fish prone to infectious diseases [2,3,4]. In order to contribute to aquaculture sustainability, practices to help fish maintain optimal gastrointestinal functionality are essential [5].

Fish gut mucosa is a very active immunological site and plays an essential role in host health, directly interacting with the aquatic environment [6,7]. Fish gut-associated lymphoid tissue (GALT) involves lymphocytes, plasmatic cells, granulocytes, and macrophages [8]. This mucosal surface supports large microbial populations, which play a key role in the intestinal environment and host-microbial interactions. Fish intestinal microbiome comprises complex communities with demonstrated impact on host health, mucosal development and cellular differentiation, metabolism, nutrition, and disease resistance [9].

Interestingly, the intestinal microbiota is strongly influenced by the rearing environment and seasonal or diet changes, which could lead to host immune-related pathologies [8,10]. Nowadays, modulation of the microbiota represents an interesting alternative for enhancing fish health status with probiotics receiving great attention as dietary supplements [11,12].

Probiotics are defined as live microbial cells that confer health benefits to the host [13,14,15] and factors such as viability may regulate their effects [6,16]. *Shewanella putrefaciens* Pdp11 is a Gram Negative bacteria isolated by our research group from the skin of *Sparus aurata* [17], and it has been demonstrated as a probiotic for farmed species such as *S. aurata* and *S. senegalensis*. Dietary administration of Pdp11 alive cells promotes the growth and improves the efficiency of feed utilization [18,19,20], stimulates *S. senegalensis* [17] and *S. aurata* [21] immune systems and intestinal functionality [18], increases resistance against bacterial pathogens such as *Vibrio harveyi* and *Photobacterium damselae* subsp. *piscicida* [19,22], and improves the stress tolerance of *S. senegalensis* specimens to high stocking density [23,24]. In addition, other studies have demonstrated the ability of Pdp11 to modulate the intestinal microbiota of larval and juvenile *S. senegelensis* specimens [25,26] and to reduce the negative effects derived from the treatment with oxytetracycline, inducing the up-regulation of genes related to antiapoptotic effects and oxidative stress regulation [27].

Despite the wide range of benefits described for probiotics, concerns mainly related to safety, stability, and standardization have been raised with live probiotic cells [28]. Moreover, non-viable probiotic microorganisms have also shown ability to exert beneficial effects such as host immune stimulation [29,30] and modulation of the intestinal environment [31]. In this context, Tayernity and Guglielmetti (2011) [32] introduced the term “paraprobiotics” to name non-viable microbial cells or crude cell extracts that induce beneficial effects on the host. Heat-treatment is one of the most common methods for bacterial inactivation. In previous studies, heat inactivated Pdp11 probiotic cells showed immunostimulatory ability for *S. aurata* under in vitro and in vivo conditions [33,34]. However, as mentioned, this probiotic has demonstrated benefits for Senegalese sole when it is dietary incorporated as whole living cells, but its paraprobiotic effects have not yet been described in Senegalese sole. For this reason, the present tentative study contributes novel information on the potential use of Pdp11 as paraprobiotic in Senegalese sole. Though benefits derived from paraprobiotics have been described in homeotherm hosts [35,36], its application in aquaculture is still very limited.

Paraprobiotics have attractive advantages for industrial use. Besides the lack of potential virulence reversion, they entail decreased or no interaction with other components of the food products. Moreover, paraprobiotics supply greater food processing easiness, as they can be added before thermal processes, retaining their activity to the level required for the intended health benefits [35]. Moreover, in terms of simplicity, paraprobiotics could result in longer shelf life and greater convenience for storage and administration as well as supplements to immune-compromised individuals [37,38].

In order to evaluate the potential use of Pdp11 as a paraprobiotic in aquaculture, the effects of heat-inactivated Pdp11 cells on the intestinal microbiota, immune response, and oxidative stress were analyzed in *S. senegalensis* in comparison with effects exerted by live cells.

## 2. Materials and Methods

### 2.1. Ethics Approval

All procedures involving fish were conducted in strict accordance with Guidelines established by the European Union (2010/63/UE) and the Spanish legislation (RD 1201/2005 and law 32/2007) for the use of laboratory animals. All experiments were approved by the Ethic Committee of Animal’s Welfare of the Spanish Institute of Oceanography of Santander (CEBA-IEO).

### 2.2. Probiotic Microorganism

*Shewanella putrefaciens* Pdp11 was isolated from the skin mucus of healthy cultured gilthead seabream (*Sparus aurata*) and selected by its in vitro ability to inhibit the main pathogens of *S. senegalensis* [22]. Furthermore, this probiotic has increased resistance against bacterial infection in Senegalese sole [17,19] and enhanced growth and improved intestinal integrity in sole juveniles [18,19]. Pdp11 cells were cultured following the methodology previously described by Tapia-Paniagua et al. (2014a) [23]. Briefly, Pdp11 was cultured on tryptic soy agar (Oxoid Ltd., Basingstoke, UK) added with NaCl (1.5%) (TSAs) at 22 °C, 48 h. After incubation, the bacterial growth on the surface of all the plates was scrapped, suspended in sterile phosphate-buffered saline (PBS) (pH 7.4), and pooled. Then, cells were recovered by centrifugation (6000× *g*, 15 min, 4 °C) and the pellet was suspended in PBS, adjusted to 10^11^ colony forming units mL^−1^ (CFU mL^−1^) (O.D.600 nm = 1.5) and viable cell concentration determined by plate count on TSAs (Appendix A).

### 2.3. Experimental Diets

The commercial pellet diet Europa Elite LE-2 (Protein: 57%; Fat: 18%; Ash: 11.5%; Cellulose. 0.2%; P total: 1.7%; Skretting, Burgos, Spain) was used as basal diet. To prepare experimental diets, 20 mL of the probiotic bacterial suspension previously described was divided into two aliquots. One of them was used as live cells while the other was incubated at 60 °C for 1 h for bacterial heat-inactivation. Finally, absence of bacterial growth was checked by inoculation of an aliquot of the heat-inactivated suspension on TSAs plates and incubation at 22 °C for 48 h. Commercial feed was ground up, mixed with 10 mL of the probiotic suspension (live or dead cells; diets P and I, respectively) previously described, to obtain a dose equivalent to 10^9^ CFU g^−1^ of feed. Control diet (diet C) was processed in the same manner and added with the same volume of PBS. Finally, diets were again made into pellets, allowed to dry and stored at 4 °C until use (Appendix A). Probiotic cell suspensions and diets were prepared and viability checked at the beginning of each week during the feeding trial.

### 2.4. Experimental Design and Sample Collection

*Solea senegalensis* specimens used in this study come from natural laying of the breeder stock kept in seawater 14,000 L tanks at the Spanish Oceanographic Institute in Santander (Spain). Once the spawn was collected and its appropriate hatching index verified, larvae were cultured in 150 L seawater tanks (35.4 gL^−1^ salinity and 19 ± 0.50 °C) with a feeding based in rotifers, phytoplankton and *Artemia* enriched with Origreen (Skretting, Burgos, Spain). Weaning was carried out with Gemma microencapsulation feed (Skretting) and post larvae were transferred to 150 L seawater tanks (35.4 gL^−1^ salinity and 19 ± 0.50 °C) and fed with commercial pellet Gemma (Skretting, Burgos, Spain). Then, fry were fed 8 times a day with the commercial Europa (Skretting, Burgos, Spain), not registering any mortality episode. Once they reached 10 g body weight, they were grown in 500 L seawater tanks (35.4 gL^−1^ salinity and 19 ± 0.50 °C) at a stock density always below 7 kg m^−2^ and with a renovation rate of 300% day^−1^. Feeding during fattening was carried out with Europa feed (Skretting, Burgos, Spain), which was maintained during the experiment. From this stock, 150 specimens were randomly sampled and distributed in six 150 L seawater tanks and acclimatized for two weeks prior to the experimental period keeping the same environmental, stock density and feeding conditions previously detailed. Specimens were acclimatized for 2 weeks prior to the experimental period. Health status of fish was visually checked based on the normal swimming and feeding behavior as well as coloration of skin and gills. Additionally, one fish per tank was euthanized by clove oil overdose (200 ppm) and dissected. The spleen, liver, and kidney were sampled and cultured on TSAs plates, a bacteriological medium widely used for the determination of most common bacterial pathogens for marine fish including *Vibrio harveyi*, *Listonella anguillarum*, *Photobacteium damselae* subsp. *piscicida* [17,22,39]. Inoculated plates were incubated at 22 °C, up to 5 days and absence of bacterial growth was observed in all cases. After the acclimatization period, in which fish received commercial diet (diet C), experimental diets (control diet supplemented with live, diet P, and inactivated probiotic cells, diet I) were randomly assigned to duplicate groups. The initial fish weight was used to adjust the daily ration of feed according to the normal pattern in this species (20 g kg^−1^) and fish were fed 8 times a day for 45 days with the corresponding experimental diet. For growth parameter determinations, the specimens from each dietary treatment were weighed at 0, 15, 30, and 45 days post feeding assay. For biometric determinations, all specimens were anaesthetized with clove oil (10–20 ppm).

Prior to sampling, 45 days after the feeding trial, fish were starved for 24 h to eliminate food remains, and six fish from each tank (N = 12 per diet) were sacrificed by an overdose of clove oil. Whole intestines were obtained after careful dissection under sterile conditions and fragments (0.5 cm) of the anterior (close to the stomach) and posterior (before the rectum) intestine stored at −80 °C for gene expression and intestinal microbiota analysis.

### 2.5. RNA Isolation and Gene Expression Analysis

Total RNA was isolated from anterior and posterior intestinal samples using GeneJET RNA Purification Kit (Thermo Scientific, Madrid, Spain) according to manufacturer’s instructions. The purified RNA was suspended in DEPC-water (Sigma, St. Louis, MA, USA), quantified by using Qubit RNA BR assay kits (Thermo Scientific, Madrid, Spain) and quality checked by gel electrophoresis. DNase I treatment (Thermo Scientific, Madrid, Spain) was carried out to ensure complete DNA removal. Reverse transcription (RT) was performed using First Strand cDNA Synthesis Kit (Thermo Scientific, Madrid, Spain) with 1 µg of total RNA and using oligo(dT)18 primers. The resulting cDNA was stored at −20 °C until use.

Primers used to detect transcription of genes encoding two pro-inflammatory interleukins *(il-1β* and *il-6*), tumor necrosis factor alpha (*tnf-α*), complement components (*c3* and *c7*), lysozyme C (*lys-c1*), cluster of differentiation 8 alpha and 4 (*cd8α* and *cd4*), and heat shock proteins (*gp96*, *hsp90aa*, *hsp90ab* and *hsp70*) are listed in Table 1. In order to obtain accurate results, qPCR efficiency ((E = (10 ^[−1/slope]^ − 1) × 100) with 0.95–1 values was checked to ensure optimized and reproducible results.

Reactions for each individual intestinal samples were carried out in 10 µL final-volume consisting of 5 µL SsoAdvanced SYBR Green Supermix (Bio-Rad Laboratories, Hercules, CA, USA), 0.5 µL of each primer set (10 µM), 1 µL template cDNA, and 3 µL DEPC-water (Sigma). Quantitative PCR determinations were performed in triplicate in 96-well PCR plates and run in the CFX96 Touch Real-Time PCR Detection System (Bio-Rad Laboratories, Hercules, CA, USA). Amplification was followed by a standard melting curve from 65 °C to 95 °C, in increments of 0.5 °C for 5 s at each step, to verify the amplification of a single product. All individual intestinal samples were run in triplicate and in parallel with two reference genes, β–actin (*actb2*) and ribosomal proteins S4 (*rps4*) [40]. To assess the stability of reference genes the Genorm algorithm was used [45]. Relative mRNA expression was calculated using the comparative Cq method 2^(−∆∆Cq)^ [46].

### 2.6. DNA Extraction and Microbiota Characterization

Intestines from six specimens from each tank (N = 12 per diet) were collected and cut longitudinally on a sterile petri dish, differentiating anterior and posterior sections. In order to characterize diet effects on the intestinal microbiota, intestinal lumen contents were pooled and stored at −80 °C until DNA extraction. In this way, variation between individual fish can be represented as a whole since pooled intestinal content samples were obtained from individuals fed with the same diet. This practice was carried out in order to avoid bias, as has been proposed in previous studies [47,48,49] Total DNA from each pool was obtained as described by Martínez et al. (1998) [50]. The dried DNA pellet was resuspended in 100 μL of DECP water (Sigma). DNA was re-precipitated with sodium acetate and washed with ethanol. Finally, DNA was centrifuged (12,000× *g*, 5 min) and the pellet dried and resuspended in 100 μL DECP water (Sigma). Agarose gel (1.5%, *w/v*) electrophoresis was used with RedSafe Nucleic Acid Staining Solution 20.000X (InTRON Biotechnology, Seoul, Korea) to visually check DNA integrity. Suitable samples of each treatment were mixed and DNA quantity determined using Qubit DNA BR assay kits and the Qubit 2.0 (Thermo Scientific). Bacterial 16S rRNA V3-V4 regions were sequenced using the Illumina MiSeq Platform (Chunlab, Inc., Seoul, South Korea) with primers 341F CCTACGGGNGGCWGCAG and 805R GACTACHVGGGTATCTAATCC (ChunLab). All Illumina reads were analyzed with FastaQC software in order to assess sequence quality. Further data processing including trimming and 16S rRNA analysis and visualization were performed with pipeline based on the MOTHUR software package (1.39.5 version). Briefly, chimeras were removed by using UCHIME version 4.2 (http://www.drive5.com/usearch/manual/uchime_algo.html (accessed on 15 January 2021), effective Tags obtained) and sequences were aligned against EzBiocloud database [2018 may] and clustered into operational taxonomic units (OTUs) with 97% identity cutoff and a threshold set at 0.005% [51].

### 2.7. Growth Performance Calculations

The survival rate (SR), weight gain rate (WGR) and specific growth rate (SGR) were used to evaluate the fish survival and growth performance, respectively. Parameters were calculated as follows.
SR (%) = 100 × Nf/Ni
WGR (%) = 100 × (Wf − Wi)/Wi
SGR (%/d) = 100 × (ln Wf − ln Wi)/d

Nf: final fish number; Ni: initial fish number; Wf: final individual weight; Wi: initial individual weight; d: length of feeding trial (days).

### 2.8. Statistical Analysis

Statistical analyses were conducted using IBM SPSS Statistics 22.0. Normality and homogeneity of variance of the data was determined by using Shapiro–Wilk and Levene’s test, respectively. Differences were statistically analyzed by one-way analysis of variance (ANOVA) with Tukey and Games–Howell post hoc tests when statistical requirements were fulfilled. Non-normally distributed data were analyzed by the non-parametric Kruskal–Wallis test, followed by a multiple comparison test. Statistical significance was set for *p* ≤ 0.05.

## 3. Results

### 3.1. Growth Performance

Tanks were checked three times a day for the presence of non-consumed feed at the bottom. Fish showed similar behavior towards the diets regardless of the presence or ab-sence of the probiotics and no learning period was implemented. No mortality was observed during the feeding trial, with 100% SR in all fish groups (Table 2). Fish fed with diet P exhibited significant increased rates (*p* < 0.05) in the final body weight, WGR and SGR compared to those receiving control (diet C) and heat-inactivated probiotic (diet I) diets. On the contrary, no differences were observed in growth performance parameters between diet C and diet I (Table 2).

### 3.2. Gene Expression Analysis

No differences in relative transcription of genes involved in cellular stress *hsp90aa* and *hsp70* were observed in fish receiving diets supplemented with Pdp11 probiotic cells, whether live or inactivated, compared to fish fed with the control diet. However, after feeding with diets formulated with the probiotics, significant *gp96* and *hsp90ab* down-regulation was observed in posterior and both intestinal sections, respectively, of fish receiving diet P and diet I, compared to the control diet group (Figure 1A,B).

Genes related to the inflammatory response including *tnf-α*, *il-6* and *il-1β*, did not show significant differences in anterior intestinal sections based on the diet used. In addition, no differences were observed in the posterior intestinal sections except for *il-1β*, which resulted significantly up-regulated in fish fed with diet P compared to the control group (Figure 2A,B). On the contrary, both probiotic diets induced significant *cd4*, and *cd8α* increased transcription in anterior intestinal sections, whereas significant *cd4* down regulation was observed in the posterior intestine. Transcription of *cd8α* gene was also significantly down-regulated in posterior intestinal sections of fish fed with diet I (Figure 2).

When genes involved in the humoral immune response (*c3*, *c7* and *lys-c1*) were analyzed, changes associated with the diet were only detected in *c3* transcription. Thus, statistically significant *c3* down-regulation was observed in anterior and posterior intestinal sections of fish fed with inactivated probiotic cells, whereas significant changes were only observed in posterior sections of fish fed with diet P. On the other hand, no changes in *c7* and *lys-c1* relative gene expression were observed after probiotic diet supplementation (Figure 3A,B).

### 3.3. Analysis of the Intestinal Microbiota

After sequencing, a total of 333,384 raw reads were obtained for both forward and reverse directions. The mean read depth per sample was 55,564 ± 26,101.58 (mean ± SD). After removing non-specific amplicons, amplicons not assigned to the target taxon and chimeras, 48,330.33 ± 24,467.21 (mean ± SD) sequences were retained. Filtering based on rarefaction curves was carried out and a total of 20,723 reads per sample was obtained. Singletons and doubletons were removed and 2628 OTUs at 97% gene similarity cutoff against EzBioCloud database were used for subsequent analysis. Sequence libraries displayed Good’s coverage estimations close to 100% (97.04 ± 1.07), suggesting that the sequences identified represent most of the bacterial composition present in each sample.

Since the effects of pooling intestinal samples rely on how representative such inoculum is regarding the intestinal ecosystem, the abundance and the variety of bacterial species can be observed as a whole in any case. In this way, results obtained showed differences in the microbiota composition between fish fed with different diets (Figure 4, Figure 5 and Figure 6). Microbiota taxonomic analysis showed three dominant phyla shared by all the samples: Proteobacteria, Tenericutes, and Spirochaetes (Figure 4). After inclusion of live Pdp11 cells in the diet, increased relative abundance percentages of phyla such as Bacteroidetes, Proteobacteria, and Firmicutes and a reduction of the phylum Tenericutes were observed compared to fish receiving control diet (Figure 4A). On the other hand, fish fed with diet I showed increased Spirochaetes and decreased Tenericutes relative abundance compared to fish fed with control diet (Figure 4A). When considering microbiota composition in the posterior intestinal sections, decreased relative abundance of Spirochaetes in favor of Tenericutes was observed in fish receiving diet P, whilst the opposite was detected when diet I was administered (Figure 4B).

When microbiota composition was analyzed at family level, the microbiota of the anterior intestinal sections of *S. senegalensis* specimens fed with diet P contained a total of 19 families with relative abundance percentages >1%, and a clear dominance of *Pseudomonadaceae* and *Xanthomonadaceae* (Figure 5A). On the contrary, microbiota of fish fed with diets C and I were similar in family composition, though increased *Brevinemataceae* and decreased *Mycoplasmataceae* percentages were observed in fish fed with inactivated probiotic cells (Figure 5A). In addition, *Vibrionaceae* family was only detected in the microbiota of *S. senegalensis* specimens fed with diet I. The microbiota of posterior intestinal sections showed lower family numbers in fish fed all the diets (Figure 5B). In the case of diet P, *Mycoplasmataceae_f1* was the most abundant family (54.33%), followed by *Spirochaetaceae* (25.78%), whilst *Brevinemataceae* was more abundant in fish receiving diet C and diet I. Furthermore, *Vibrionaceae* relative percentage was higher (10.6%) in the microbiota of fish receiving diet I. On the contrary, *Spirochaetaceae* was detected in the microbiota of posterior intestinal sections of fish fed control diet and diet P. Appendix A shows relative abundance percentages at family level.

Dominant genera comprising the microbiota in all samples (diet P, diet I and control groups) are represented in Figure 6 (relative abundance percentages >1%). Administration of diet P noticeably increased the number of genera and phylotypes in anterior intestinal sections, with *Pseudomonas*, *Bacteroides*, and *Stenotrophomonas* genera to the head (Figure 6A). On the contrary, this variety decreased in posterior intestinal sections, with only four genera (*Brevinema*, *Mycoplasma_g12*, *Pseudomonas* and *Stenotrophomonas*) and two phylotypes (*Mycoplasmataceae_f1* and AF166259_g (*Spirochaetaceae*)) showing relative abundance percentages above 1% (Figure 6B). Among them, *Brevinema* genus and AF166259_g phylotype represented about 80% of the total abundance (Figure 6B). The dietary inclusion of inactivated probiotic cells resulted in increased abundance of genera such as *Vibrio* and especially *Brevinema*, in both anterior and posterior intestinal regions, with very high dominance in the latter, where these genera represented almost 88% of the total sequences (Figure 6A,B). A remarkable absence of *Mycoplasma_g12* genus was observed in the posterior sections of fish fed the inactivated cells (diet I) compared to microbiota of fish receiving diet C and diet P (Figure 6B). Overall, a total of 231 and 263 genera were identified in the anterior and posterior intestinal sections, respectively (Appendix A). Of these, 27 and 36 genera were shared by the microbiota of anterior and posterior intestinal sections, respectively, regardless of the diet received by the fish (Appendix A). Microbiota of specimens fed with the diet supplemented with live probiotic cells showed higher number of unique genera (71 and 85 in anterior and posterior intestinal sections, respectively) compared to fish fed with control or inactivated probiotic diet. This latter, showed higher abundance of unique genera in anterior intestinal sections (60) compared to fish fed control diet (34) (Appendix A).

## 4. Discussion

The improvements in growth performance in fish fed with some heat-killed probiotics such as *Lactobacillus plantarum* strain L-137 in *Oreochromis niloticus* [52] or *Bacillus sp*. SJ-10 in *Paralichthys olivaceus* [53] compared to basal diets have been reported. In the case of the probiotics *S. putrefaciens* Pdp11, benefits associated to the administration of viable cells are well documented in *S. aurata* and *S. senegalensis* [24,27,54,55]. However, advantages in terms of safety and product stability derived from the use of non-viable cells make the investigation of the effects of inactivated cells worthwhile. Results obtained in the present work indicate that dietary administration of heat-inactivated Pdp11 cells to *S. senegalensis* does not result in similar growth performance improvement compared to that observed with live cells, though no decreased growth was detected compared to the control diet.

Several authors have reported that probiotics supplemented in the fish diet can effectively modulate the intestinal microbiota, with associated improved growth, immunity, and disease resistance [56]. In agreement with results previously reported by Tapia-Paniagua et al., 2010 [24], Spirochaetes and Proteobacteria were predominant phyla in the intestine of *S. senegelansis* specimens. Interestingly, Tenericutes considerably increased in the anterior sections of fish fed diet C and I, but, except for fish receiving probiotic live cells, it was practically absent in the posterior sections (Figure 4). In this context, Bano et al. (2007) [57], suggest that different Mycoplasma ribotypes may dominate in anterior and posterior intestine, suggesting partitioning by location in the digestive tract of the fish *Gillichthys mirabilis*. This could explain differences in the bacterial taxa abundances in both intestinal sections detected in this study, which could be related to enzymatic activities or slightly to pH values along the gut tract of *S. senegalensis* [58].

On the other hand, in this study, the highest bacterial diversity was observed in the microbiota of fish fed with the diet P. Bacteroidetes are represented by a series of philotypes belonging to *Muribaculaceae* family (PAC000186_g, PAC001074_g and PAC001112_g) and *Ruminococcaceae* family (PAC000661_g). Both, Bacteroidetes and Firmicutes are associated with better amino acid, carbohydrate, and lipid metabolism in *Penaeus monodon* [59]. However, *Bacteroidaceae* family is one of the predominant taxa in anterior intestinal sections of fish fed the diet P. In this sense, *Bacteroides* genus has been reported as member of the intestinal microbiota of freshwater and marine fish [60,61,62] with the ability to transform the primary bile acids [63] in essential metabolic regulators for lipid metabolism [64]. In this context, previous studies [19,25,26], have shown significant differences in the fatty acid profiles and the presence of lipid inclusions in the liver of *S. senegalensis* fed with a diet supplemented with Pdp11 compared to specimens receiving a control diet. These findings point out to a better metabolic activity for fish fed the diet P.

The analysis at genus level in anterior and posterior intestinal samples showed the predominance of genus *Brevinema*, especially in fish fed control and inactivated diets. This genus has also been previously reported as predominant in *S. senegalensis* intestinal tract [25,26]. High abundance of the phylum Spirochaetes has also been observed in carnivorous fish microbiota, including mahi-mahi (*Coryphaena hippurus*) and great barracuda (*Sphyraena barracuda*) [65]. In addition, Corrigan et al. (2015) [66] found that Spirochaetes, jointly with Firmicutes, could play an important role in the fermentation of dietary carbohydrates, transporting non-digestible sugars across fish cellular membranes. Normally, there is a divergence in the capacity to utilize dietary carbohydrates between carnivorous species, which cannot be satisfactory due to anomalies in glucose metabolism [67]. In this sense, the great abundance of Spirochaetes and therefore of the genus *Brevinema*, mostly induced by the probiotic inactivated cells, could be related to promoting a better ability to deal with glucose in a fish species, that, despite its carnivorous feeding habits, seems to have high requirements in this term [68].

Furthermore, a common phylum in marine fish is Proteobacteria [69], standing out for participating in various biogeochemical processes (carbon, nitrogen, and sulfur cycling) in aquatic ecosystems [70]. Here, this phylum is present in the intestinal microbiota of fish receiving all diets and is mostly represented by genera including *Pseudomonas*, *Stenotrophomonas*, *Luteibacter*, and *Vibrio*. It should be noted that, unlike in fish receiving inactivated probiotic cells, *Vibrio* suffers a reduction in fish fed with the P diet, resulting in agreement with previous studies [27]. Bacterial diseases occur easily when the immunity of fish declined, or the diet and environment changed. In any case, in our study, *Vibrio* seems not to play a pathogenic role according to the absence of fish mortality and disease symptoms as well as in the gene expression results obtained.

Gastrointestinal microbes play a critical role in the development and maturation of GALT, which in turn mediate a variety of host immune functions [71]. As a consequence, diets modulating the intestinal microbiota, may play important roles in mediating immunity [72]. In this work, the immunomodulatory effect on *S. senegalensis* caused by dietary probiotic administration was studied. Immunomodulation by intestinal microbiota is attributed to the effects on the release of cytokines, including interleukins, tumor necrosis factors, among others, which regulate both innate and adaptive immune response. Heat shock proteins (HSPs) play a key role in the cell stress response system, preventing protein aggregation, promoting protein homeostasis and it can be up-regulated in cells exposed to different abiotic and biotic factors [44,73,74]. Two important isoforms (*hsp90aa* and *hsp90ab*) proteins have been found to exert protective effects against heat or cold shock, hyperosmotic stress, immune response, and heavy metal toxicity and have essential roles in the folding and translocation of proteins [75]. Protein encoded by *gp96* is also involved in the immune response, including maturation of antigen presenting cells and stimulation of pro-inflammatory cytokines [76]. Notable increased transcription in *gp96* gene expression, and associated pro-inflammatory response, has been reported in *S. senegalensis* specimens infected with *P. damselae* subsp. *piscicida* [77]. Similarly, *Saprolegnia parasitica* uptake process by host cells is guided by a gp96-like receptor [78]. On the other hand, stimulation of *hsp70* and *hsp90* gene expression is also considered as a marker to assess stress [79]. In this context, similar relative *hsp70*, *hsp90* expression in fish fed with both experimental diets (P and I), jointly with *gp96* and *hsp90ab* down-regulation, could indicate that *S. senegalensis* specimens fed with the probiotic diets were not under a situation involving cellular stress in the intestinal environment.

Additionally, increased expression of pro-inflammatory genes is indicative of an immunologically activated status [80]. In fish, the involvement of *il-6* in the promotion of antibody production has been reported in Nile tilapia (*Oreochromis niloticus*) [80] and Japanese pufferfish (*Fugu rubripes*) [81]. Furthermore, in agreement with previous studies, the supplementation of live and heat-inactivated probiotic feed additives increased the expression of pro-inflammatory genes, including *il-1β* and *tnf-α*, normally enhancing the phagocytic activities of leukocytes [82,83,84]. Similarly, feeding with Pdp11 live or dead cells has resulted in increased *il-1β* in the posterior intestinal sections. Considering the fact that pro-inflammatory cytokines may boost immune response, but prolonged and excessive expression might result in negative effects, results observed could suggest an immune support in fish, especially when live cells are administered.

On the other hand, complement components play an important role in non-specific and specific immunity [85]. In fish, the complement system is the major component of the innate immune response, and plays an essential role in microbial killing, phagocytosis, inflammatory reactions, immune complex clearance, and antibody production [86]. Several studies demonstrated that probiotics such as *Pediococcus acidilactici*, *Lactobacillus rhamnosus*, *Enterococcus faecium*, and *Lactobacillus plantarum* modulate *c3* and *c4* gene expression associated with protection against bacterial pathogens in fish [87]. In this way, probiotic administration could improve complement component levels in fish when facing infectious stress. In this way, increased complement gene expression was not observed in the present study and a general and similar down-regulation of complement genes was detected regardless of the probiotic diet administered. Normally, components of the complement system increase their synthesis immediately upon inflammatory stimuli and improve fish immunity [85]. However, a tight regulation of inflammatory processes is necessary to avoid adverse effects of immune activation, especially in the absence of the pathogen [88]. Similarly, lysozyme (*lys-c1*) is a conserved antimicrobial protein critical in host defense against pathogens [89], and like the majority of gene expression levels observed, could indicate that individuals are not under stress.

Several studies documented improved immune response after incorporation of probiotics and paraprobiotics in fish diets as results of the interaction between bacterial and intestinal cells [90]. However, this interaction can occasionally result in dysbiosis. This is a basic type of cellular stress and a common reason for hsp induction [91,92]. Genes such as *gp96* and *hsp90ab*, have been involved in the acute phase response and the immune regulation against microbial pathogens [93,94,95]. Dysbiosis has also been associated with increased transcription of pro-inflammatory cytokines [96,97]. In our study, changes of the intestinal microbiota observed in fish fed the diets P and I were neither associated to significant increased transcription of genes encoding key proteins in the stress response nor pro-inflammatory cytokines, suggesting that dysbiosis was not induced. Furthermore, this phenomenon is also associated to T cell imbalance and inflammation [98]. In our study, fish receiving the diets supplemented with viable or inactivated probiotic cells showed significant increases of *cd4* and *cd8α* genes in the anterior intestinal sections. The direct killing activity by an apparent MHC-independent mechanism against both facultative intracellular and extracellular bacteria demonstrated by CD4 and CD8α cells has been reported [99], and it could explain the up-regulation detected in this study without inflammatory response. In this way, this up-regulation does not imply killing activity to occur in all cases because it is dependent on the effector and target levels [99], and it could be in agreement with the absence of inflammatory and oxidative stress response observed in our study.

## 5. Conclusions

In this work, the effects of heat-inactivated cells of the probiotic strain Pdp11 on the intestinal microbiota and immune response were evaluated in *S. senegalensis*. Results obtained show that the probiotic dietary inclusion of live or inactivated cells modulate the microbiota and expression of genes involved in the intestinal immune response of *S. senegalensis*. Similar relative gene transcription levels were obtained in fish fed both diets, showing that bacterial viability is not essential for the immunomodulatory effect observed in *S. senegalensis*. On the contrary, remarkable differential modulation of the intestinal microbiota, with higher variety of bacterial taxa, was observed when fish received live probiotic cells. Notwithstanding, further research including controlled intervention studies with challenge assays will contribute to improving the information on the potential effects of diets during infection with pathogenic microorganisms. In addition, mechanistic studies will allow the dilucidation of the mechanisms involved in the different response to the diets and ascertain other potential benefits on *S. senegalensis*.

## Figures and Tables

**Figure 1 microorganisms-09-00808-f001:**
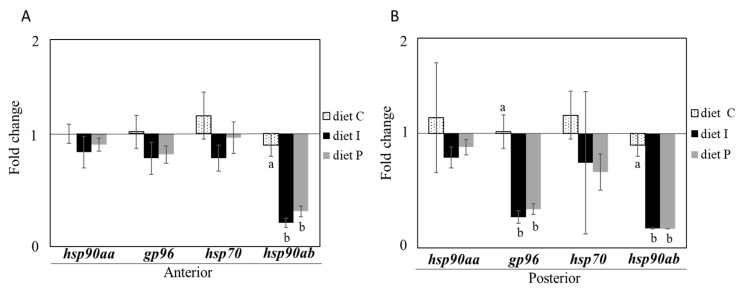
Relative transcription (normalized with *actb2* and *rps4*) of genes involved in cellular stress in anterior (**A**) and posterior (**B**) intestinal sections of *S. senegalensis* fed with live (diet P) and inactivated (diet I) Pdp11 cells. Four genes including *hsp90aa*, *gp96*, *hsp70* and *hsp90ab* were analyzed by RT-qPCR. Data (mean ± standard deviation) are presented as relative fold change of target genes in fish fed with supplemented diets compared to control group (diet C). Different letters indicate significant differences between diets determined by one-way ANOVA test (*p* ≤ 0.05).

**Figure 2 microorganisms-09-00808-f002:**
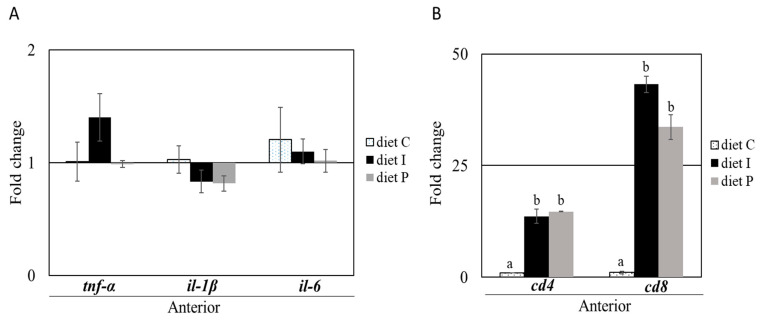
Relative gene transcription (normalized with *actb2* and *rps4*) of genes involved in pro-inflammatory and immune response in anterior (**A**,**B**) and posterior (**C**) intestinal sections of *S. senegalensis* fed with live (diet P) and inactivated (diet I) Pdp11 cells. Five genes including *tnf-α*, *il-1β*, *il-6*, *cd8α*, and *cd4* were analysed by RT-qPCR. Data (mean ± standard deviation) are presented as relative fold change of target genes in fish fed with supplemented diets compared to control group (diet C). Different letters indicate significant differences between diets determined by one-way ANOVA test (*p* ≤ 0.05).

**Figure 3 microorganisms-09-00808-f003:**
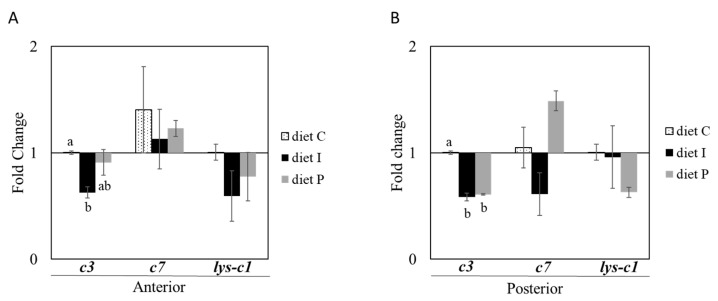
Relative gene transcription (normalized with *actb2* and *rps4*) of genes involved in complement components and lysozyme in anterior (**A**) and posterior (**B**) intestinal sections of *S. senegalensis* fed with alive (diet P) and inactivated (diet I) cells of Pdp11. Three genes including *c7*, *c3*, and *lys-c1* were analysed by RT-qPCR. Data (mean ± standard deviation) are presented as relative fold change of target genes in fish fed with supplemented diets compared to control group (diet C). Different letters indicate significant differences between diets determined by one-way ANOVA test (*p* ≤ 0.05).

**Figure 4 microorganisms-09-00808-f004:**
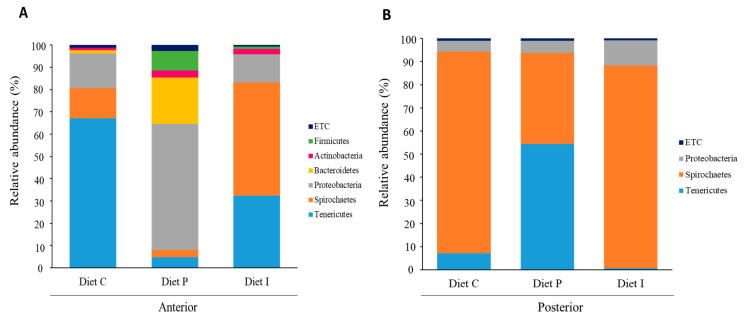
Microbiota composition (relative operational taxonomic units (OTUs) composition) at phylum level of pooled intestinal contents from anterior (**A**) and posterior (**B**) intestinal sections of *S. senegalensis* specimens fed with control diet (diet C) and diet supplemented with live (diet P) or inactivated (diet I) Pdp11 cells. ETC indicates relative abundance percentage below 1%.

**Figure 5 microorganisms-09-00808-f005:**
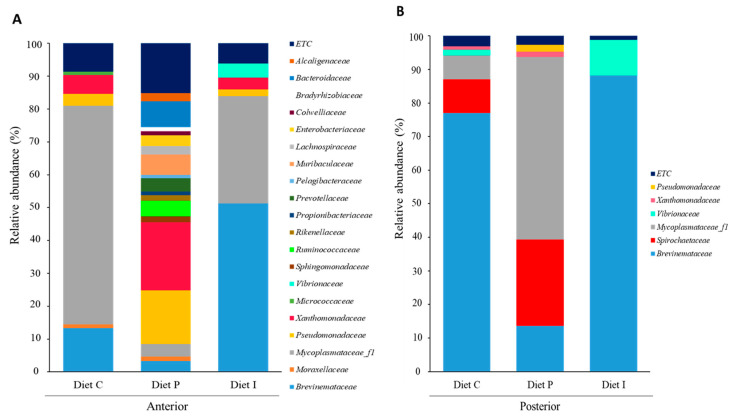
Microbiota composition (relative OTUs composition) at family level of pooled intestinal contents from anterior (**A**) and posterior (**B**) intestinal sections of *S. senegalensis* specimens fed with control diet (diet C) and diets supplemented with live (diet P) or inactivated (diet I) Pdp11 cells. ETC indicates relative abundance percentage below 1%.

**Figure 6 microorganisms-09-00808-f006:**
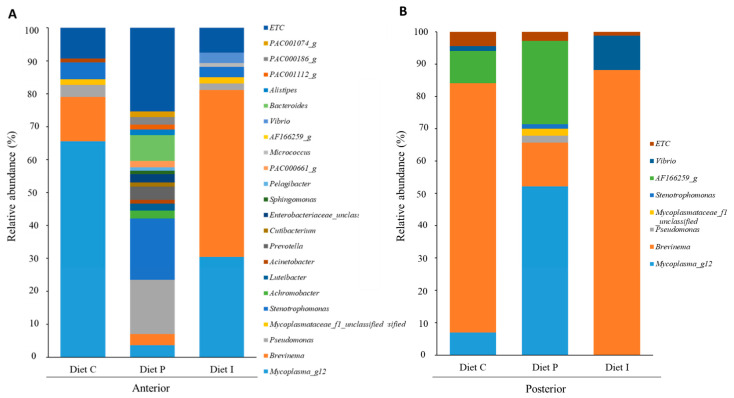
Microbiota composition (relative OTUs composition) at genus level of pooled intestinal contents from anterior (**A**) and posterior (**B**) intestinal sections of *S. senegalensis* specimens fed with control diet (diet C) and diets supplemented with live (diet P) or inactivated (diet I) Pdp11 cells. ETC indicates relative abundance percentage below 1%.

**Table 1 microorganisms-09-00808-t001:** List of *S. senegalensis* genes analyzed in this study.

Gene	Code	Sequence	Reference
Beta actin 2	*actb2*	AATCGTGACCTCTGCTTCCCCCTGT (F) TCTGGCACCCCATGTTACCCCATC (R)	[40]
Ribosomal protein S4	*rps4*	GTGAAGAAGCTCCTTGTCGGCACCA (F) AGGGGGTCGGGGTAGCGGATG (R)	[40]
Interleukin 1 beta	*il-1* *β*	CGCAGAAAGTGACATGTTGAGATTT (F) GGAAGCGGGCAGACATGA (R)	[41]
Interleukin 6	*il-6*	ACAATTTCCTGCAGAGATAAAAGTAAGCT (F) CAAGCCCTCAGGCCTACAATATTAA (R)	[41]
Complement C3	*c3*	ACCTTAGACTGCCCTACTCTGCTGTCCGTG (F) GCACTGCACACATCATCCGTCTCAGAC (R)	[42]
Complement C7	*c7*	GGCACACACTATCTGTCGCAGGGCTC (F) GGCGAACGCCTGATGGTTTAACTCCAG (R)	[42]
C1 type lysozyme	*lys-c1*	CAGATCAACAGCCGCTATTGG (F) GCTGATTCCACATGCATTTGAAGTG (R)	[41]
Cluster of differentiation 4	*cd4*	GACCTCAGGCTGCAATGGT (F) TGAGCAGAGTGATGGACAGACT (R)	[41]
Cluster of differentiation 8 alpha	*cd8* *α*	GTGCCAGCATTAAAAGCAACGA (F) GCAGTCACAACTTCCGCTCTTT (R)	[41]
Tumor necrosis factor alpha	*tnfα*	CTGGGACTGCTGGCACTTGGATTTG (F) CAGTTCTCCACGCTGACGTACTGTCGAAC (R)	[43]
Heat shock protein GP96	*gp96*	GAGTCTTCTCCCTTTGTTGAGCGGCTG (F) TGATGCCTTCCTTTGCCACGTTCTG (R)	[43]
Heat shock protein 90AA	*hsp90aa*	GACCAAGCCTATCTGGACCCGCAAC (F) TTGACAGCCAGGTGGTCCTCCCAGT (R)	[44]
Heat shock protein 90AB	*hsp90ab*	TCAGTTTGGTGTGGGTTTCTACTCGGCTTA (F) GCCAAGGGGCTCACCTGTGTCG (R)	[44]
Heat shock protein 70	*hsp70*	GCTATACCAGGGAGGGATGGAAGGAGGG (F) CGACCTCCTCAATATTTGGGCCAGCA (R)	[43]

**Table 2 microorganisms-09-00808-t002:** Growth performance of *S. senegalensis* fed three different diets over 45 days. The three diets were: control diet (Diet C), the same diet supplemented with alive probiotic cells (Diet P), and inactivated probiotic cells (Diet I).

	Diet C	Diet P	Diet I
Initial body weight (g)	28.52 ± 1.42 ^a^	30.18 ± 1.37 ^a^	30.87 ± 1.37 ^a^
Final body weight (g)	34.86 ± 2.44 ^a^	46.40 ± 2.18 ^b^	37.10 ± 2.18 ^a^
SR (%)	100.00 ± 0.00 ^a^	100.00 ± 0.00 ^a^	100.00 ± 0.00 ^a^
WGR (%)	22.23 ± 3.35 ^a^	48.81 ± 4.12 ^b^	20.18 ± 3.01 ^a^
SGR (%/d)	0.45 ± 0.02 ^a^	0.88 ± 0.05 ^b^	0.41 ± 0.02 ^a^

Values are means ± standard deviations of three replicate groups. Mean values with the different superscripts in the same row are significantly different (*p* < 0.05). SR: survival rate; WGR: weight gain rate; SGR: specific growth rate.

## Data Availability

Molecular sequence data reported in this paper were deposited in the National Center for Biotechnology Information (NCBI) Sequence Read Archive (SRA accession: PRJNA694873).

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
