# Peer review of "A Tentative Study of the Effects of Heat-Inactivation of the Probiotic Strain Shewanella putrefaciens Pdp11 on Senegalese Sole (Solea senegalensis) Intestinal Microbiota and Immune Response"

_microorganisms, 2021, doi:10.3390/microorganisms9040808_

Round 1
Reviewer 1 Report
What was the rationale for heat-killing the bacterial culture as opposed to using other methods of cell inactivation?
What were the scientific reasons for adopting a 45-day feeding regime? Do the authors have data showing that 45-days is the optimum period to apply supplemented feed?
Do you have any challenge data that could be added to the manuscript, i.e. is there any evidence for protection against pathogens? I would encourage the authors to obtain challenge data is they do not already have some available.
TITLE: Include the common name of Solea senegalensis in the title
METHODS, 2.2. It would be useful for the lay reader if the authors could add some more information about the source of pdp11
METHODS, 2.3. 109 CFU?
METHODS, 2.4. How did you check the health status of the fish?
RESULTS, 3.1. It would be helpful to add a comment about the response of the fish to the diets – did the fish consume the diets as well as or better than the controls?
Was there a learning period for the fish to become used to the diets?
Was there any difference in behaviour/vigour/health among the different groups?
Author Response
Manuscript ID: microorganisms-1117250
Type of manuscript: Article
Title: Effects of heat-inactivation of the probiotic strain Shewanella putrefaciens Ppd11 on Solea senegalensis intestinal microbiota and immune response
Dear Mr. Wade Wang,
We are thankful to the editor and reviewers for considering our manuscript and appreciate their comments, which undoubtedly contribute to improving the manuscript. Below we detail the answers to their suggestions.
Yours sincerely,
Marta Domínguez Maqueda
- Reviewer 1
What was the rationale for heat-killing the bacterial culture as opposed to using other methods of cell inactivation?
Heat-treatment is one of the most common methods for bacterial inactivation. In previous studies, heat-inactivated probiotic cells showed immunostimulatory ability for Sparus aurata under in vitro and in vivo conditions (Díaz-Rosales et al., 2006; Salinas et al., 2006). In the present work, we aimed to study the effects of heat-inactivated probiotics dietary inclusion on the modulation of both immune response and intestinal microbiota in Senegalese sole.
This information has been specified in the manuscript (Introduction, lines 74-76).
“Heat-treatment is one of the most common methods for bacterial inactivation. In previuos studies, heat inactivated probiotic cells showed immunostimulatory ability for Sparus au-rata under in vitro and in vivo conditions [33,34].”
What were the scientific reasons for adopting a 45-day feeding regime? Do the authors have data showing that 45-days is the optimum period to apply supplemented feed?
Most studies carried out on juveniles of both, Senegalese sole and gilthead seabream, have adopted feeding periods comprised between 28 and 69 days (Cámara-Ruiz et al., 2020). In addition, it is commonly considered that a period in which fish experience a considerable weight gain is appropriate to evaluate most effects of supplemented feed.
Finally, a 45-day feeding period has been demonstrated to be able to modulate intestinal microbiota in Senegalese sole specimens fed with a diet supplemented with the macroalgae Ulva ohnoi (Tapia-Paniagua et al., 2019). For this reasons, a 45-day period was considered to study the effects of the heat-inactivated probiotic cells in the present work.
Cámara-Ruiz, et al. 2020. Microorganisms, 8(12), 1990. https://doi.org/10.3390/microorganisms8121990
Tapia-Paniagua et al., 2019. Frontiers in Microbiology, 10, 171. doi.org/10.3389/fmicb.2019.00171
Do you have any challenge data that could be added to the manuscript, i.e. is there any evidence for protection against pathogens? I would encourage the authors to obtain challenge data is they do not already have some available.
Evidence of protection against several bacterial pathogens including Vibrio harveyi, Listonella anguillarum and Photobacterium damselae subsp. piscicida has been obtained for alive Pdp11 cells (Chabrillón et al., 2005, 2006; Díaz-Rosales et al., 2009). In all the cases, Pdp11 dietary incorporation reduced fish mortality. Appropriate installations for pathogen manipulation were not available at the time of the present feeding trial and no infection assays could be carried out. However, considering that modulation of the intestinal microbiota and immune response has been reported with alive Pdp11 probiotic cells (Tapia-Paniagua et al., 2014, 2015; Jurado et al., 2018), the aim of this work was the evaluation of the effects of the inactivated cells in comparison to alive cells.
Chabrillón et al., 2005. Journal of Fish Diseases, 28, 229-237. doi.org/10.1111/j.1365-2761.2005.00623.x
Chabrillón et al., 2006. Aquaculture Research, 37(1), 78-86. doi.org/10.1111/j.1365-2109.2005.01400.x
Díaz Rosales et al., 2009. Aquaculture, 293, 16-21. doi.org/10.1016/j.aquaculture.2009.03.050
Tapia-Paniagua et al., 2014. Fish & Shellfish Immunology, 41, 209-221. doi.org/10.1016/j.fsi.2014.08.019
Tapia-Paniagua et al., 2015. Fish & Shellfish Immunology, 46, 449-458. doi.org/10.1016/j.fsi.2015.07.007
Jurado et al., 2018. Fish & Shellfish Immunology, 77, 350-363. doi.org/10.1016/j.fsi.2018.04.018
TITLE: Include the common name of Solea senegalensis in the title.
According to the accurate reviewer’s suggestion, the common name has been included.
METHODS, 2.2. It would be useful for the lay reader if the authors could add some more information about the source of pdp11
Information about the probiotic strain Pdp11 has been included in “2.2. Probiotic microorganism” (Materials and methods, section 2.2, lines 99-103).
“Shewanella putrefaciens Pdp11 was isolated from the skin mucus of healthy cultured gilthead seabream (Sparus aurata) and selected by its in vitro ability to inhibit the main pathogens of S. senegalensis[22]. Furthermore, this probiotic has in-creased resistance against bacterial infection in Senegalese sole [17,19] and enhanced growth and improved intestinal integrity in sole juveniles[18,19].”
METHODS, 2.3. 109 CFU?
The right sentence is “Commercial feed was ground up, mixed with 10mL of the probiotic suspension (alive or dead cells; diets P and I, respectively) previously described, to obtain a dose equivalent to 109 CFU g-1 of feed”. This has been corrected in the manuscript (Materials and methods, section 2.3, lines 127-129).
METHODS, 2.4. How did you check the health status of the fish?
Health status of fish was visually checked based on the normal swimming and feeding behaviour and coloration of skin and gills In addition, in order to check for the absence of pathogen infections, microbiological analysis of internal organs was carried out on tryptic soy agar (Oxoid Ltd., Basingstoke, UK) supplemented with 1.5% NaCl (TSAs). This bacteriological medium is widely used for the determination of most common bacterial pathogens for marine fish, including Vibrio harveyi, Listonella anguillarum, Photobacteium, damselae subsp. piscicida (Chabrillón et al., 2005, 2006; Díaz-Rosales et al., 2009). Inoculated plates were incubated at 22ºC, up to 5 days and absence of bacterial growth was observed in all cases.
Chabrillón et al., 2005. Journal of Fish Diseases, 28, 229-237. doi.org/10.1111/j.1365-2761.2005.00623.x
Chabrillón et al., 2006. Aquaculture Research, 37(1), 78-86. doi.org/10.1111/j.1365-2109.2005.01400.x
Díaz Rosales et al., 2009. Aquaculture, 293, 16-21. doi.org/10.1016/j.aquaculture.2009.03.050
Part of this information had already been included in the manuscript, but it has been modified in order to make it clearer (Materials and methods, section 2.4, lines 161-168).
“Health status of fish was visually checked based on the normal swimming and feeding behaviour well as coloration of skin and gills. Additionally, one fish per tank was eu-thanized by clove oil overdose (200 ppm) and dissected out. The spleen, liver and kidney were sampled and cultured on on TSAs plates, a bacteriological medium widely used for the determination of most common bacterial pathogens for marine fish including Vibrio harveyi, Listonella anguillarum, Photobacteium, damselae subsp.piscicida [17,22,39]. Inoculated plates were incubated at 22ºC, up to 5 days and absence of bacterial growth was observed in all cases.”
RESULTS, 3.1. It would be helpful to add a comment about the response of the fish to the diets – did the fish consume the diets as well as or better than the controls?
Tanks were checked three times a day for the presence of non-consumed feed at the bottom. Fish showed similar behaviour towards the diets regardless of the supplementation or absence of the probiotics. This information has been included in the manuscript (Results, section 3.1, lines 260-262).
“Tanks were checked three times a day for the presence of non-consumed feed at the bottom. Fish showed similar behaviour towards the diets regardless of the presence or absence of the probiotics and no learning period was implemented.”
Was there a learning period for the fish to become used to the diets?
After the acclimatization period with commercial pellet diet (diet C), experimental diets (control diet supplemented with the probiotic cells) were administered. This way, farmed fish did not have an abrupt change in the diet. As mentioned before, fish showed similar behaviour towards the diets regardless of the presence or absence of the probiotics.
The comments of the reviewer have been considered in Materials and methods, section 2.2, lines 166-168:
“After the acclimatization period, in which fish received commercial diet (diet C), experimental diets (control diet supplemented with alive, diet P, and inactivated probiotic cells, diet I) were randomly assigned to duplicate groups.”
Was there any difference in behaviour/vigour/health among the different groups?
No mortality was observed during the feeding trial, and behaviour and health did not show differences. On the other hand, differences were observed in the final body weight, weight gain rate (WGR) and specific growth rate (SGR) between fish fed with the diet containing alive probiotic cells compared to control and inactivated probiotic diets. This information has been included in the manuscript (Results, section 3.1, lines 262-267).
“No mortality was observed during the feeding trial, with 100% SR in all fish groups (Table 2). Fish fed with diet P exhibited significant increased rates (p<0.05) in the final body weight, WGR and SGR compared to those receiving control (diet C) and heat-inactivated probiotic (diet I) diets. On the contrary, no differences were observed in growth performance parameters between diet C and diet I (Table 2).”

Reviewer 2 Report
This study fully studies the effects of dietary administration of heat-inactivated cells of the probiotic strain Shewanella putrefaciens (as a parabiotic) on the the intestinal microbiota and immune gene transcription in Solea senegalensis. The data obtained revealed hsp90ab, gp96, cd4, cd8, il-1β and c3 transcription was modulated after probiotic supplementation, with no differences between viable and heat-inactivated probiotic supplemented diets. However, differences in growth and modulation of intestinal microbiota was only observed when live Shewanella putrefaciens was administered in the diet. Molecular techniques used are the most appropriate for complex microbiota analysis and the results obtained are relevant for the field and can be consider an advance in the current knowledge. This study have generated a big amount of data. Authors’ analysis and interpretation was clearly presented and most conclusions are well supported by the results.
However, I think some details should have been included or discussed, and the authors need to consider the issues listed below:
Major points
- Results section. Figure 1B. Standard deviations of hsp90aa expression with diet C and hsp70 expression with diet I are extremely high in comparison with the other samples and could have an effect in the final conclusions, as the authors state that there are not differences in hsp90aa or hsp70 expression between the different diets. I wonder if RT-qPCR should have been repeated with those samples. Same thing happens in Fig 3 with standard deviation with c7 gene expression in diet C
Minor points:
Line 97. Please put 9 as a superscript in 109: 109
Line 101. TSA plates, not TSAs plates. Please correct throughout the manuscript.
Line 309. Brevinema should be in italics
Line 355. Please rephrase this sentence “the diversity in bacterial taxa observed in the microbiota of fish fed with the P diet is an evidence”
Author Response
Manuscript ID: microorganisms-1117250
Type of manuscript: Article
Title: Effects of heat-inactivation of the probiotic strain Shewanella putrefaciens Ppd11 on Solea senegalensis intestinal microbiota and immune response
Dear Mr. Wade Wang,
We are thankful to the editor and reviewers for considering our manuscript and appreciate their comments, which undoubtedly contribute to improving the manuscript. Below we detail the answers to their suggestions.
Yours sincerely,
Marta Domínguez Maqueda
- Reviewer 2
This study fully studies the effects of dietary administration of heat-inactivated cells of the probiotic strain Shewanella putrefaciens (as a parabiotic) on the the intestinal microbiota and immune gene transcription in Solea senegalensis. The data obtained revealed hsp90ab, gp96, cd4, cd8, il-1β and c3 transcription was modulated after probiotic supplementation, with no differences between viable and heat-inactivated probiotic supplemented diets. However, differences in growth and modulation of intestinal microbiota was only observed when live Shewanella putrefaciens was administered in the diet. Molecular techniques used are the most appropriate for complex microbiota analysis and the results obtained are relevant for the field and can be consider an advance in the current knowledge. This study have generated a big amount of data. Authors’ analysis and interpretation was clearly presented and most conclusions are well supported by the results.
However, I think some details should have been included or discussed, and the authors need to consider the issues listed below:
Major points
Results section. Figure 1B. Standard deviations of hsp90aa expression with diet C and hsp70 expression with diet I are extremely high in comparison with the other samples and could have an effect in the final conclusions, as the authors state that there are not differences in hsp90aa or hsp70 expression between the different diets. I wonder if RT-qPCR should have been repeated with those samples. Same thing happens in Fig 3 with standard deviation with c7 gene expression in diet C
The comments of the reviewer have been considered. The samples indicated by the reviewer have been repeated and although the standard deviations obtained are slightly lower, the values obtained still do not show significant differences between the different diets. All samples were analysed in the same way (reagents, system, protocols…) and no significant differences were observed between technical replicates.
Minor points:
Line 97. Please put 9 as a superscript in 109: 109
The right sentence is “Commercial feed was ground up, mixed with 10mL of the probiotic suspension (alive or dead cells; diets P and I, respectively) previously described, to obtain a dose equivalent to 109 CFU g-1 of feed”. This has been corrected in the manuscript (Materials and methods, section 2.3, lines 127-129)
.Line 101. TSA plates, not TSAs plates. Please correct throughout the manuscript.
We would like to point out that TSAs plates stands for tryptic soy agar (TSA) as basal medium, supplemented with NaCl (1.5%) (TSAs). TSAs is commonly used to name tryptic soy agar added with NaCl.
On the other hand, since our study is related to the marine farmed fish Senegalese sole, culture of bacteria from marine fish, pathogens or non-pathogens, needs to be carried out on TSAs plates.
Line 309. Brevinema should be in italics.
Brevinema has been italicized (Results, section 3.1, line 394).
Line 355. Please rephrase this sentence “the diversity in bacterial taxa observed in the microbiota of fish fed with the P diet is an evidence”
The comments of the reviewer have been considered. The phrase has been changed in the manuscript (Discussion, lines 441-442):
“The major bacterial diversity in the microbiota is observed on fish fed with the diet P”

Reviewer 3 Report
Shewanella putrefaciens Ppd11 – Solea senegalensis
The manuscript reports limited novel scientific information and the experimental approach is questioned.
The results obtained for Ppd11 as a potential paraprobiotic culture for Solea sengalensis(“sole”) appears not to go beyond results published in the literature and referred to in the manuscript. Like most previous investigations the experimental approach is imperical and not mechanistic to verify the paraprobiotic effects claimed in literature and in the manuscript.
Further the investigation is seen as an intervention study but strictly speaking not meeting some of the basic requirements for such studies. It should have been a double blinded, placebo controlled and randomized intervention study.
The study is not blinded.
Details are missing in Materials and Methods Section 2.3 to define “diet C” as a valid placebo and the composition of the “commercial pellet diet” is not given. Volumes of PBS, if any, added “diet C”, during the 45 days trials are not given. A figure illustrating the set-up could be useful.
Details are also missing in 2.4 to define “healthy farmed … juvenile” fish, their prehistory including acclimatization and the criteria for the fish to be included in the intervention study.
According to 2.4 a total of 150 S.s. were randomly distributed in six 150-L seawater tanks and later the three diets i.e the placebo and the diets P and I with S.s were randomly assigned to duplicate groups. Further explanation is needed to understand how randomizations should be understood and to clarify if the immune and stress related transcriptomics responses determined on day 45 were carried out on pooled samples or individual S.s. This is important to know for evaluation of the experimental and scientific scope of the study.
It is not mentioned if number of viable Ppd11 added to “diet P” and heat killed for “diet I” were checked and controlled throughout the 45 days intervention. The check for health status of S.s. described in 2.4 seems not to be appropriate, information on the specific pathogens in mind would have been useful. Is the 24 h starvation at the end of the intervention mentioned in 2.4 stressful for the fish i.e. will it involve a stress related transcriptomic response? In general sample sizes for analyses seem not to be defined.
It is expected that the composition of the microbiota vary between individual fish. However, the data available from the study appear not to indicate such expected variations between the individual fish in the tanks and diets(Figures 4, 5 and 6).
The need for mechanistic studies is mentioned in Conclusion. I the same way the need for controlled intervention studies shoudl have been mentioned should have been mentioned.
Author Response
Manuscript ID: microorganisms-1117250
Type of manuscript: Article
Title: Effects of heat-inactivation of the probiotic strain Shewanella putrefaciens Ppd11 on Solea senegalensis intestinal microbiota and immune response
Dear Mr. Wade Wang,
We are thankful to the editor and reviewers for considering our manuscript and appreciate their comments, which undoubtedly contribute to improving the manuscript. Below we detail the answers to their suggestions.
Yours sincerely,
Marta Domínguez Maqueda
- Reviewer 3
The manuscript reports limited novel scientific information and the experimental approach is questioned.
The results obtained for Ppd11 as a potential paraprobiotic culture for Solea sengalensis (“sole”) appears not to go beyond results published in the literature and referred to in the manuscript. Like most previous investigations the experimental approach is empirical and not mechanistic to verify the paraprobiotic effects claimed in literature and in the manuscript.
Up to the author’s knowledge, this is the first study reporting the effects of inactivated probiotic cell dietary administration to Senegalese sole. Previous studies have shown the in vitro effects by using Sparus aurata derived cell line (Salinas et al., 2006) and in gilthead seabream specimens (Diaz-Rosales et al., 2006). However, response to supplements and probiotics in the diet is dependent on the fish species, especially if fish have different feeding habits. The aim of the present work is to determine the effects of dietary inactivated Pdp11 cells on the intestinal microbiota and innate immune response of Senegalese sole. Though this probiotics has demonstrated benefits for Senegalese sole when it is dietary incorporated as whole cells, its paraprobiotic effects have not been described in Senegalese sole yet. For this reason, we consider that the present study contributes novel information on the potential use of Pdp11 as paraprobiotics in Senegalese sole.
We appreciate reviewer’s suggestion and, once the potential paraprobiotics effects have been demonstrated, a mechanistic approach will be considered in order to deep into the mechanisms of action of the heat-inactivated probiotic cells in further studies.
Further the investigation is seen as an intervention study but strictly speaking not meeting some of the basic requirements for such studies. It should have been a double blinded, placebo controlled and randomized intervention study.
The study is not blinded.
The study included a placebo group, which was called “control group” (diet C) and consisted on a group of fish fed with the commercial diet added with an equal volume of saline solution as the probiotic diets, but devoid of probiotics. In addition, all fish specimens used in the assay were randomly assigned to the experimental tanks and randomly sampled. Also, treatment compliance was determined by inspection and quantification of not consumed aquafeed in each tank, and all fish showed similar behaviour towards assayed diets.
We agree with the reviewer that the trial was not conceived as a double-blinded study. However, despite this is more common in clinical studies, the number of research articles published in the field of aquaculture based on studies not blinded is high (Guirro et al., 2019; Peixoto et al., 2019; Altaib et al., 2021; Reyes‐Becerril et al., 2021; Mohamed et al., 2021). In addition , researchers did not consider in any case the origin and type of sample and treated all of them similarly, regardless of the treatment group.
Altaib et al., 2021. Microorganisms, 9, 378. doi.org/10.3390/microorganisms9020378
Guirro et al., 2019. PloS one, 14(9), e0218143. doi.org/10.1371/journal.pone.0218143
Mohamed et al., 2021. Aquaculture Reports, 19, 100567. doi.org/10.1016/j.aqrep.2020.100567
Peixoto et al., 2019. Scientific Reports, 9, 16134. doi.org/10.1038/s41598-019-52693-6
Reyes‐Becerril et al., 2021. Aquaculture Research. doi.org/10.1111/are.15123
Details are missing in Materials and Methods Section 2.3 to define “diet C” as a valid placebo and the composition of the “commercial pellet diet” is not given. Volumes of PBS, if any, added “diet C”, during the 45 days trials are not given. A figure illustrating the set-up could be useful.
According to reviewer’s suggestions, detailed information on the proximate composition and preparation of diets has been incorporated to the manuscript (Materials and methods, section 2.3, lines 121-133). In addition, a Figure illustrating the set-up has been included as Supplementary material (Figure S.1).
“The commercial pellet diet Europa Elite LE-2 (Protein: 57%; Fat: 18%; Ash: 11.5%; Cellulose. 0.2%; P total: 1.7%, Skretting, Spain) was used as basal diet. To prepare experi-mental diets, 20mL of the probiotic bacterial suspension previously described was divided into two aliquots. One of them was used as alive cells while the other was incubated at 60ºC for 1 h for bacterial heat-inactivation. At last, absence of bacterial growth was checked by inoculation of an aliquot of the heat-inactivated suspension on TSAs plates and incubation at 22ºC for 48 h. Commercial feed was ground up, mixed with 10mL of the probiotic suspension (alive or dead cells; diets P and I, respectively) previously described, to obtain a dose equivalent to 109 CFU g-1 of feed . Control diet (diet C) was processed in the same manner and added with the same volume of PBS. Finally, diets were again made into pellets, allowed to dry and stored at 4°C until use (Figure S.1). Probiotic cell sus-pensions and diets were prepared and viability checked at the beginning of each week during the feeding trial.”
Details are also missing in 2.4 to define “healthy farmed … juvenile” fish, their prehistory including acclimatization and the criteria for the fish to be included in the intervention study.
According to reviewer’s suggestions, this information has been incorporated to the manuscript (Materials and methods, section 2.4, lines 141-157).
“Solea senegalensis specimens used in this study come from a natural laying of the breeder stock kept in seawater 14,000 L tanks at the Spanish Oceanographic Institute in Santander (Spain). Once the spawn was collected and its appropriate hatching index veri-fied, larvae were cultured in 150L seawater tanks (35.4 gL-1 salinity and 19±0.50°C) with a feeding based in rotifers, phytoplankton and Artemia enriched with Origreen (Skretting, Burgos, Spain). Weaning was carried out with Gemma microencapsulation feed (Skret-ting) and post larvae were transferred to 150L seawater tanks (35.4 gL-1 salinity and 19±0.50°C) and fed with commercial pellet Gemma (Skretting Burgos, Spain). Then, fry were fed 8 times a day with the commercial Europa (Skretting Burgos, Spain), not regis-tering any mortality episode. Once they reached 10 g body weight, they were grown in 500 L seawater tanks (35.4 gL-1 salinity and 19±0.50°C) at a stock density always below 7kg m-2 and with a renovation rate of 300% day-1. Feeding during fattening was carried out with Europa feed (Skretting Burgos, Spain), which was maintained during the experiment. From these stock, 150 specimens were randomly sampled and distributed in six 150L seawater tanks and acclimatized for two weeks prior the experimental period keeping the same environmental, stock density and feeding conditions previously detailed.”
According to 2.4 a total of 150 S.s. were randomly distributed in six 150-L seawater tanks and later the three diets i.e the placebo and the diets P and I with S.s were randomly assigned to duplicate groups. Further explanation is needed to understand how randomizations should be understood and to clarify if the immune and stress related transcriptomics responses determined on day 45 were carried out on pooled samples or individual S.s. This is important to know for evaluation of the experimental and scientific scope of the study.
It has been specified in the manuscript that the immune and stress related transcriptomic responses determined on day 45 were carried out on individual intestinal samples. Furthermore, information on the randomization process to allocate diets has also been included (Materials and methods, section 2.5, lines 203 and 211).
It is not mentioned if number of viable Ppd11 added to “diet P” and heat killed for “diet I” were checked and controlled throughout the 45 days intervention. The check for health status of S.s. described in 2.4 seems not to be appropriate, information on the specific pathogens in mind would have been useful. Is the 24 h starvation at the end of the intervention mentioned in 2.4 stressful for the fish i.e. will it involve a stress related transcriptomic response? In general sample sizes for analyses seem not to be defined.
According to reviewer’s suggestions, details on the preparation of diets and number of viable Pdp11 cells have been incorporated to the manuscript (sections 2.2 and 2.3respectively). In addition, diets were prepared at the beginning of each week during the feeding trial, and both viability and absence of bacterial growth checked for probiotic and heat-inactivated diets, respectively, throughout the 45 days intervention. This information has been specified in the manuscript (Materials and methods, section 2.2, lines 13-110). In addition, a Figure illustrating the set-up has been included as Supplementary material (Figure S.1).
“Pdp11 cells were cultured following the methodology previously described by Tapia-Paniagua et al. (2014a) [23]. Briefly, Pdp11 was cultured on tryptic soy agar (Oxoid Ltd., Basingstoke, UK) added with NaCl (1.5%) (TSAs) at 22ºC, 48h. After incubation, the bacterial growth on the surface of the all plates was scrapped, suspended in sterile phos-phate-buffered saline (pH 7.4), and pooled. Then, cells were recovered by centrifugation (6000 xg, 15min, 4ºC) and the pellet was suspended in PBS, adjusted to 1011 colony form-ing units mL-1 (CFU mL−1) (O.D.600nm = 1.5) and viable cell concentration determined by plate count on TSAs (Figure S.1).”
Health status of fish was visually checked based on the normal swimming and feeding behaviour and coloration of skin and gills In addition, in order to check for the absence of pathogen infections, microbiological analysis of internal organs was carried out on tryptic soy agar (Oxoid Ltd., Basingstoke, UK) supplemented with 1.5% NaCl (TSAs). This bacteriological medium is widely used for the determination of most common bacterial pathogens for marine fish, including Vibrio harveyi, Listonella anguillarum, Photobacteium, damselae subsp. piscicida (Chabrillón et al., 2005, 2006; Díaz-Rosales et al., 2009). Inoculated plates were incubated at 22ºC, up to 5 days and absence of bacterial growth was observed in all cases.
Chabrillón et al., 2005. Journal of Fish Diseases, 28, 229-237. doi.org/10.1111/j.1365-2761.2005.00623.xChabrillón et al., 2006. Aquaculture Research, 37(1), 78-86. doi.org/10.1111/j.1365-2109.2005.01400.x
Díaz Rosales et al., 2009. Aquaculture, 293, 16-21. doi.org/10.1016/j.aquaculture.2009.03.050
Part of this information had already been included in the manuscript, but it has been modified in order to make it clearer (Materials and methods, section 2.4, lines 161-168).
“Health status of fish was visually checked based on the normal swimming and feeding behaviourras well as coloration of skin and gills. Additionally, one fish per tank was eu-thanized by clove oil overdose (200 ppm) and dissected out. The spleen, liver and kidney were sampled and cultured on on TSAs plates, a bacteriological medium widely used for the determination of most common bacterial pathogens for marine fish including Vibrio harveyi, Listonella anguillarum, Photobacteium, damselae subsp.piscicida [17,22,39]. Inoculated plates were incubated at 22ºC, up to 5 days and absence of bacterial growth was observed in all cases. ”
A 24h-starvation period is widely practiced in aquaculture, especially in studies related to farmed fish and the investigation of the effects of nutritional treatments, infections, immune parameters (Rufchaei et al., 2020; Makled et al., 2020; Guardiola et al., 2018; Tachibana et al., 2020; Mohammadian et al., 2019). The fasting condition reduces the fish metabolic rate and decreases stress. In addition, a 24h-starvation period contributes to a digestive tract devoid of the presence of feces, and therefore, allow the study of gut autochthonous microbiota of fish associated to the different diets administered.
Guardiola et al., 2018. Fish & Shellfish Immunology, 74, 372-379. https://doi.org/10.1016/j.fsi.2018.01.010
Makled et al., 2020. Probiotics and antimicrobial proteins, 12(2), 365-374. doi.org/10.1007/s12602-019-09575-0
Mohammadian et al., 2019. Fish & Shellfish Immunology, 86, 269-279. doi.org/10.1016/j.fsi.2018.11.052
Rufchaei et al., 2020. Aquaculture, 515, 734533. doi.org/10.1016/j.aquaculture.2019.734533
Tachibana et al., 2020. Aquaculture Reports, 16, 100277. doi.org/10.1016/j.aqrep.2020.100277
It is expected that the composition of the microbiota vary between individual fish. However, the data available from the study appear not to indicate such expected variations between the individual fish in the tanks and diets (Figures 4, 5 and 6).
In this study, variation between individual fish is represented as a whole since pooled intestine samples were obtained from individuals fed with the same diet. This practice was carried out in order to avoid bias, as it has been proposed in previous studies (Wu et al., 2010; Hao et al., 2017; Tapia-Paniagua et al., 2020). The effects of pooling intestinal samples rely on how representative such inoculum is regarding the intestinal ecosystem, but the abundance and the variety of bacterial species can be observed in any case. In this way, results obtained showed differences in the microbiota composition between fish fed with different diets.
Wu et al., 2010. Aquaculture 303:1–7. doi.org/10.1016/j.aquaculture.2009.12.025
Hao et al., 2017. Probiotics and Antimicrobial Proteins, 9, 386–396. doi.org/10.1007/s12602-017-9269-7
Tapia-Paniagua et al., 2020. Aquaculture Nutrition. 1–11. DOI: 10.1111/anu.13039
Németh et al., 2003. European journal of nutrition, 42(1), 29-42. doi.org/10.1016/j.mimet.2014.08.022
The need for mechanistic studies is mentioned in Conclusion. I the same way the need for controlled intervention studies should have been mentioned.
According to reviewer’s suggestions, this information has been incorporated to the manuscript (Conclusions, section 5, lines 560-565).
“Notwithstanding, further research including controlled intervention studies with chal-lenge assays will contribute to improving the information on the potential effects of diets during infection with pathogenic microorgansims. In addition, mechanistic studies will allow to dilucidate the mechanisms involved in the different response to the diets and as-certain other potential benefits on S. senegalensis.”

Round 2
Reviewer 3 Report
The comments given to the authors have been addressed to the stage that publication is recommended. However, further editing to improve readability and clarity should be considered. The concern expressed in the comments concerning "individual" vs."pooled" samples should be kept in mind for the data behind Figures 4, 5 and 6 and addressed in "Materials and Methods". As mentioned in the original Review Report the study is regarded as an intervention study but not meeting some basic requirements for such studies. Therefore it is recommended that the overall title be added "- A tentative study" and the reason be explained in "Introduction".
Author Response
Manuscript ID: microorganisms-1117250
Type of manuscript: Article
Title: Effects of heat-inactivation of the probiotic strain Shewanella putrefaciens Ppd11 on Solea senegalensis intestinal microbiota and immune response
Dear Mr. Wade Wang,
We are thankful to the editor and reviewers for considering our manuscript and appreciate their comments, which undoubtedly contribute to improving the manuscript. Below we detail the answers to their suggestions.
Yours sincerely,
Marta Domínguez Maqueda
Round 2
- Reviewer 3
The comments given to the authors have been addressed to the stage that publication is recommended. However, further editing to improve readability and clarity should be considered. The concern expressed in the comments concerning "individual" vs."pooled" samples should be kept in mind for the data behind Figures 4, 5 and 6 and addressed in "Materials and Methods". As mentioned in the original Review Report the study is regarded as an intervention study but not meeting some basic requirements for such studies. Therefore it is recommended that the overall title be added "- A tentative study" and the reason be explained in "Introduction".
According to the previous report and reviewer’s new suggestions, detailed information concerning intestinal samples has been incorporated to the manuscript (with track changes under the name of “Reviewer 3. Round2”) in the different sections and subsections:
Materials and methods, section 2.6, lines 225-228:
“In this way, variation between individual fish is represented as a whole since pooled intestinal content samples were obtained from individuals fed with the same diet. This practice was carried out in order to avoid bias, as it has been proposed in previous studies (Wu et al., 2010; Hao et al., 2017; Tapia-Paniagua et al., 2020).”
Wu et al., 2010. Aquaculture 303:1–7. doi.org/10.1016/j.aquaculture.2009.12.025
Hao et al., 2017. Probiotics and Antimicrobial Proteins, 9, 386–396. doi.org/10.1007/s12602-017-9269-7
Tapia-Paniagua et al., 2020. Aquaculture Nutrition. 1–11. DOI: 10.1111/anu.13039
Results, section 3.1, lines 357-361:
“Since the effects of pooling intestinal samples rely on how representative such inoc-ulum is regarding the intestinal ecosystem, the abundance and the variety of bacterial species can be observed as a whole in any case. In this way, results obtained showed differences in the microbiota composition between fish fed with different diets (Figures 4, 5, 6).”
In addition, it has been specified at the bottom of the figures 4, 5 and 6 that intestinal samples were pooled:
Results. Section 3.1. Lines 374-376.
“Figure 4. Microbiota composition (relative OTUs composition) at phylum level of pooled intestinal contents from intestinal contents from anterior (A) and posterior (B) intestinal sections of S. senegalensis specimens fed with control diet (diet C) and diet supplemented with alive (diet P) or inactivated (diet I) Pdp11 cells. ETC indicates relative abundance percentage below 1%.”
Results. Section 3.1. Lines 397-399.
“Figure 5.- Microbiota composition (relative OTUs composition) at family level of pooled intestinal contents from anterior (A) and posterior (B) intestinal sections of S. senegalensis specimens fed with control diet (diet C) and diet supplemented with alive (diet P) or inactivated (diet I) Pdp11 cells. ETC indicates relative abundance percentage below 1%.”
Results. Section 3.1. Lines 427-429.
“Figure 6.- Microbiota composition (relative OTUs composition) at genus level of pooled intestinal contents from intestinal contents from anterior (A) and posterior (B) intestinal sections of S. senegalensis specimens fed with control diet (diet C) and diets supplemented with alive (diet P) or inactivated (diet I) Pdp11 cells. ETC indicates relative abundance percentage below 1%.”
Also, the overall title has been changed:
“A tentative study of the effects of heat-inactivation of the probiotic strain Shewanella putrefaciens Ppd11 on Senegalese sole (Solea senegalensis) intestinal microbiota and immune response”
Finally, it has been more clearly specified that this study is considered a tentative study in the Introduction section (lines 79-83):
“However, as mentioned, this probiotic has demonstrated benefits for Senegalese sole when it is dietary incorporated as whole living cells, but its paraprobiotic effects have not been described in Senegalese sole yet. For this reason, the present tentative study contributes novel information on the potential use of Pdp11 as paraprobiotic in Senegalese sole.”